# Interleukin 24: Signal Transduction Pathways

**DOI:** 10.3390/cancers15133365

**Published:** 2023-06-27

**Authors:** Simira Smith, Sual Lopez, Anastassiya Kim, Justina Kasteri, Ezekiel Olumuyide, Kristian Punu, Columba de la Parra, Moira Sauane

**Affiliations:** 1Department of Biological Sciences, Herbert H. Lehman College, City University of New York, 250 Bedford Park Boulevard West, Bronx, NY 10468, USA; simira.smith@lc.cuny.edu (S.S.); sual.lopez@lc.cuny.edu (S.L.); justina.kasteri@lc.cuny.edu (J.K.); ezekiel.olumuyide@lc.cuny.edu (E.O.); kristian.punu@lc.cuny.edu (K.P.); 2Ph.D. Program in Biology, The Graduate Center, City University of New York, 365 Fifth Avenue, New York, NY 10016, USA; akim1@gradcenter.cuny.edu (A.K.); columba.delaparra@lehman.cuny.edu (C.d.l.P.); 3Department of Chemistry, Herbert H. Lehman College, City University of New York, 250 Bedford Park Boulevard West, Bronx, NY 10468, USA

**Keywords:** interleukin 24, interleukin 20, interleukin 10, eIF4F complex, PKA, PKR, PERK, eIF2 alpha, translation regulation, p38 MAPK, ROS, UTR, tumor necrosis factor-alpha, ceramide, JAK/STAT, Grim 19, sigma 1 receptor, BiP/GRP78

## Abstract

**Simple Summary:**

Interleukin 24 is a pleiotropic immunomodulatory cytokine. Numerous studies have shown that enhancing or inhibiting the expression of Interleukin 24 has a beneficial effect in animal models and clinical trials in different pathologies. We and others have already demonstrated the therapeutic utility of Interleukin 24 as an anticancer therapy and in autoimmune diseases and inflammation. Successful drug targeting will require a deeper understanding of the downstream signaling pathways. In this review, we discuss the signaling pathway triggered by Interleukin 24.

**Abstract:**

Interleukin 24 is a member of the IL-10 family with crucial roles in antitumor, wound healing responses, host defense, immune regulation, and inflammation. Interleukin 24 is produced by both immune and nonimmune cells. Its canonical pathway relies on recognition and interaction with specific Interleukin 20 receptors in the plasma membrane and subsequent cytoplasmic Janus protein tyrosine kinases (JAK)/signal transducer and activator of the transcription (STAT) activation. The identification of noncanonical JAK/STAT-independent signaling pathways downstream of IL-24 relies on the interaction of IL-24 with protein kinase R in the cytosol, respiratory chain proteins in the inner mitochondrial membrane, and chaperones such as Sigma 1 Receptor in the endoplasmic reticulum. Numerous studies have shown that enhancing or inhibiting the expression of Interleukin 24 has a therapeutic effect in animal models and clinical trials in different pathologies. Successful drug targeting will require a deeper understanding of the downstream signaling pathways. In this review, we discuss the signaling pathway triggered by IL-24.

## 1. Introduction

Interleukin (IL) 24 belongs to the IL-10 cytokine family, classified into three subfamilies: the IL-10 subfamily (consisting of IL-10 itself only), the IL-20 subfamily (comprising IL-19, IL-20, IL-22, IL-24, and IL-26), and type III interferons (IFNs; comprising IL-28A/IFN-λ2, IL-28B/IFN-λ3, and IL-29/IFN-λ1) [1,2,3]. IL-24 is produced by immune (T, B, NK, innate lymphoid and dendritic cells, monocytes, and macrophages [4]) and nonimmune cells (melanocytes, dermal keratinocytes, IL-1 stimulated human colonic subepithelial myofibroblasts, and epithelial stem cells at wound edges [5,6]). IL-24 binds to and signals through heterodimers of the IL-20 receptor subunit B (IL-20RB) and either the IL-20 receptor subunit A (IL-20RA) or the IL-22 receptor subunit A1 (IL-22RA1). The binding of IL-24 to its receptors activates the Janus kinase (JAK)/signal transducer and activator of the transcription (STAT) pathway within the cytoplasm. Although the expression of IL-24 receptors is limited to specific tissues, in cancer cells, the expression of IL-24 receptors is widely expressed [6], including in melanoma, prostate, pancreatic, fibrosarcoma, ovarian, and breast cancer.

In recent years, it has been shown that IL-24 interacts with chaperones in the endoplasmic reticulum, binding immunoglobulin protein (BiP) and sigma 1 receptor (σ1) when IL-24 triggers apoptosis in cancer cells. It was recently discovered that IL-24 interacts with a gene associated with retinoid-IFN-induced Mortality 19 (Grim19), a respiratory chain protein in the inner mitochondrial membrane in T cells [7], promoting the recruitment of STAT3 to mitochondria. IL-24 binds to dsRNA-activated protein kinase (PKR), leading to apoptosis in lung cancer cells [8]. Misfolded IL-24 accumulation in the cytoplasm of patients with proteasome-associated autoinflammatory syndrome leads to the binding and activation of PKR, causing innate inflammation [9]. Therefore, IL-24 can bind not only to its classic protein plasma membrane receptors, the IL-20 Receptors [10], but also to other intracellular molecular partners, including BiP/GRP78 [11], σ1 [12], Grim19 [7], and PKR [8]. 

## 2. Canonical IL-24 Signaling through Binding to IL-20 Receptor and JAK/STAT Pathway Activation Depends on Concentration

Like all IL-10 family members, IL-24 signals via the JAK/STAT signaling pathway after ligand–receptor interaction in the cell membrane (Figure 1). IL-24 activates JAK1, JAK3, and Tyk2, as well as STAT1 and STAT3 [13,14]. Dumoutier et al. were the first to report that IL-24 binds to the IL-20 heterodimeric receptor complex comprising IL-20R1/IL-20R2 and to the IL-22 complex, comprising IL-22R/IL-20R2, resulting in the activation of STAT3 signaling pathways in epithelial cells [10,14]. Parrish-Novak et al. showed that lower doses of IL-24 induce proliferation in receptor-transfected BaF3 cells, whereas high doses of IL-24 bind to IL-20R, which leads to the growth inhibitory effect in an ovarian carcinoma cell line; they demonstrated that a similar growth inhibitory effect was observed with IL-19 but not IL-20 in a human ovarian carcinoma cell line [13]. They discussed the possibility that different IL-20 receptor affinities or the binding of IL-24 with an unknown receptor could explain the difference between IL-24 and IL-19 compared with IL-20 ligands [13]. Their findings emphasize the relevance of a quantitative understanding of IL-24 action.

In cancer cells, we were the first to show that IL-24 induces apoptosis independently of the JAK/STAT pathway in melanoma, breast, fibrosarcoma, and prostate cancer cell lines [15]. Later, Chada et al. confirmed that IL-24 induces apoptosis in melanoma cell lines independently of the STAT3 mechanism and demonstrated that a high concentration of IL-24 can induce apoptosis via an IL-20 receptor-dependent mechanism in melanoma cell lines [16]. We demonstrated that nonsecreted intracellular IL-24 protein induced apoptosis in prostate cancer cell lines [17]. Therefore, there is a clear distinction between the signaling pathways triggered by a high concentration of IL-24 followed by induction of apoptosis in cancer cells via a JAK/STAT-independent mechanism and the canonical JAK/STAT signaling pathway activation by physiological concentrations of IL-24.

## 3. SOCS1/3 Pathway

Cytokine signaling is negatively regulated by a family of suppressors of cytokine signaling proteins (SOCS). SOCS extinguish the cytokine-induced JAK/STAT pathway [18,19,20]. This resolution of cytokine receptor signaling by SOCS plays an essential regulatory function in homeostasis [21] (Figure 1). This negative feedback loop triggered by SOCS proteins is essential to avoid the progression of inflammatory diseases or cancer. 

IL-24 activates SOCS3 proteins through JAK/STAT-dependent pathways to limit astrocyte inflammatory responses following bacterial infection [22]. Chong et al. showed that IL-24 repressed the proinflammatory Th17 cytokine program, limiting the pathogenicity of Th17 cells via SOCS1/3 activation [23]. IL-17A inhibits the Th17 cytokine program via negative feedback regulation through autocrine induction of IL-24, preventing the Th17 pathogenicity in autoimmune diseases. IL-24-induced SOCS1 and SOCS3 limited the Th17 pathogenicity by inhibiting Th17 lineage cytokines, such as IL-17, GM-CSF, and probably IL-22 [23]. These discoveries open a new therapeutic possibility for various inflammatory and autoimmune diseases, such as multiple sclerosis, rheumatoid arthritis, Crohn’s disease, and autoimmune uveitis; increased IL-24 expression in addition to IL-17A could be a possible therapeutic strategy. 

IL-24 activates SOCS protein in the inflamed mucosa of patients with inflammatory bowel disease: IL-24 is induced in the mucosa by IL-1β in human colonic subepithelial myofibroblasts [24] through IL-24 mRNA stabilization via p38 MAPK activation once IL-24 is induced, and it increases the expression of membrane-bound mucins by inducing SOCS3 protein without affecting the proliferation or expression of proinflammatory cytokines such as IL-8, IL-6, or TNFα (Figure 2). Therefore, SOCS proteins play an important downstream role in the IL-24 signal transduction pathway for various inflammatory and autoimmune diseases. Further experiments are needed to determine if SOCS protein plays a role in IL-24-dependent cancer apoptosis. Based on the signal transduction pathways that IL-24 can activate in cancer cells, it is possible to speculate that IL-24 could induce the expression of a more stable and active truncated isoform of SOCS3.

SOCS proteins are induced via the JAK/STAT pathway and JAK/STAT-independent pathways via TNFα, PKA, and p38 MAPK. Interestingly, activation of SOCS proteins also occurs through JAK/STAT-independent pathways, such as TNFα, the activity of PKA, and p38 MAPK [25,26,27] (see below and Figure 2). The fact that IL-24 can activate the expression of TNFα, as well as activate PKA and p38 MAPK in cancer cells, suggests that IL-24 might have the ability to activate SOCS through JAK/STAT-independent pathways, which can be relevant information in the context of cancer treatment and immune modulatory response. 

## 4. p38 MAPK Pathway

IL-24 induces activation of multiple receptor-dependent intracellular signaling components, including transient activation of mitogen-activated protein kinases (MAPK): ERK, p38, and JUN N-terminal kinase (JNK) in intestine epithelial cell lines, which might contribute to IL-24′s anti-inflammatory and protective role [24]. We have shown that IL-24 induces sustained activation of p38 MAPK activity, contributing to cancer cell apoptosis [28] (Figure 3). We also demonstrated that IL-24 stabilizes its mRNA without activating its promoter [29]; subsequent studies by Otkjaer et al. showed that p38 MAPK regulates IL-24 expression by stabilization of the 3′UTR of its mRNA [30]. IL-1β induces IL-24 expression by stabilizing IL-24 mRNA via p38 MAPK activation in human colonic subepithelial myofibroblasts [24] and, similarly, after treatment with IL-1β in keratinocytes [30].

IL-24 not only activates p38 MAPK but can also activate JNK pathways and inhibit extracellular signal-regulated kinases (ERKs) in cancer cells [31]. For example, the protective role of nuclear factor erythroid 2-related factor 2 (Nrf2) against oxidative stress was inhibited by IL-24 via p38 MAPK and JNK in cervical and lung carcinoma cell lines [31]. IL-24 is not the only member of the IL-20 family able to regulate MAPK pathways. IL-20 can also activate the p38 MAPK pathway in monocyte-derived dendritic cells [32]. IL-19 suppresses osteoclast differentiation by inhibiting p38 MAPK activation [33]. The production of IL-22 can activate p38 MAPK in a variety of cells, including intestinal epithelial cells [34], rat hepatoma cell line [35], and keratinocytes [36]. 

## 5. ROS Production

Lebedeva et al. showed for the first time that lL-24 induces apoptosis by promoting mitochondrial dysfunction and reactive oxygen species (ROS) production in prostate cancer cells [37]. Furthermore, IL-24 inhibits the translocation of Nrf2 to the nucleus, which mediates oxidative stress response. IL-24 mediates Nrf2 inhibition by activating the p38 MAPK-dependent signal pathway, which potentiates the association of Nrf2 and Keap1 and inhibits the ERK MAPK signal pathway to dilate Nrf2 nuclear translocation [31]. Overall, studies [12,17,37,38,39,40] suggest that in cancer cells, IL-24 causes ER stress, which induces the production of ROS. Conversely, in normal mouse vascular aortic smooth muscle cells, IL-24 inhibits cell growth by decreasing ROS production and enhancing the expression of antioxidant enzymes [41]. 

Interestingly, ROS inducers lead to the translation of IL-24 protein, resulting in toxicity in pancreatic cancer cells [42]. Moreover, the induction of IL-24 in human oral keratinocyte cell line HOK-16B stimulated with Tannerella forsythia is regulated by ROS, MAPKs, and IL-6 [43]. Thus, ROS inducers lead to the translation of IL-24 primarily on keratinocytes during inflammation and in pancreatic cancer cells, and reverse IL-24 induces ROS production in cancer cells. It remains unknown if IL-24 can robustly induce the expression of endogenous IL-24 by ROS production. 

## 6. PKR and PERK Pathways

Protein kinase R (PKR), a serine–threonine kinase, belongs to a family of four proteins, including (PKR)-like endoplasmic reticulum kinase (PERK), general control nonderepressible 2 kinase (GCN2), and heme-regulated eIF2α kinase (HRI), which are involved in ER stress response, nutrient deprivation, and heme-mediated translational control, respectively. All members of this family of kinases regulate protein synthesis via the eukaryotic translation initiation factor 2 (eIF2α) pathway, a translation initiation regulator. A variety of stress–response pathways regulate PKR. For example, PKR is activated by pathogen invasion (e.g., double-stranded RNA and bacterial lipopolysaccharide), various stressful cellular conditions, such as DNA damage, mechanical stress, and ER stress, energy excess, and the presence of heat shock proteins, mitochondrial RNA, cytokines (e.g., TNFα, IL-1, and IFN-γ), calcium, ROS, growth factors (e.g., PDGF), and heparin. 

IL-24 induces activation of both PKR and PERK (Figure 4). Exogenous IL-24 activates PERK during ER stress in different cancer cells [40,44,45]. PERK phosphorylates the alpha subunit of the eIF2, and induces both ATF4 and GADD153/CHOP. eIF2, in turn, is an inhibitor of translation initiation. Hence, the crosstalk between PERK, eIF2α, and ATF4/CHOP plays a crucial role in IL-24-induced apoptosis [28,46,47]. IL-24 induces apoptosis via PKR in lung and breast tumor cells [16,48]. Interestingly, recent studies have uncovered the role of PKR as an innate immune sensor activated by proteasome dysfunction. In this context, a cytoplasmic spliced variant of IL-24 binds and activates PKR under conditions of proteasome inhibition. Activation of PKR, and the consequent phosphorylation of STAT1 and eIF2α followed by an inflammatory response (IFN-αβ, TNF, IL-6), have been shown in patients with proteasome-associated autoinflammatory disease [9]. Parker et al. speculated that the IL-24 anticancer effect could be associated with IFN-αβ expression triggered by the activation of PKR by cytosolic IL-24, since IFN-αβ plays a direct effect on target cancer cells by activating immune responses [49]. For all of the above, it can be stated that PKR and PERK play a key role in apoptosis triggered by IL-24 and probably in the inflammatory response.

## 7. Chaperones—Role in Endoplasmic Reticulum Stress-Associated Pathways

We previously demonstrated that IL-24 can induce apoptosis via both secretory and nonsecretory signaling mechanisms [17]. We have shown that the adenovirus (Ad) vector expressing IL-24 (Ad.IL-24), IL-24 protein (generated from Ad.IL-24-infected cells), and adenovirus vector expressing nonsecretable IL-24 all display broad cancer-specific pro-apoptotic activity through induction of endoplasmic reticulum (ER) stress [17,39] (Figure 4). Mechanistically, both secreted and intracellular expressions of IL-24 in cancer cells mediated apoptosis through JAK/STAT-independent and p38 MAPK-dependent pathways and induced sustained ER stress as evidenced by expression of ER stress markers (BiP/GRP78, GRP94, XBP1, and eIF2α). In the ER, IL-24 interacted with two chaperones: BiP/GRP78 and Sigma 1 Receptor (σ1) [17,38]. We have shown that IL-24 interacts with σ1 and that interaction is a critical upstream signal for IL-24-induced apoptosis in cancer cells (Figure 4). Consistent with our findings, Bina and colleagues have recently shown the interaction of IL-24 and Sig1R in an in silico analysis [50]. σ1 is a ligand-regulated membrane protein that is an important drug target for treating Alzheimer’s disease, chemotherapy-induced neuropathic pain, drug abuse, psychoses, viral infection, and cancer. σ1 is found in the ER, plasma membrane, cytoplasmic membrane systems, and nuclear envelope. σ1 can bind to an extensive list of proteins, including G-protein-coupled receptors, inositol triphosphate receptor (IP3R), and ion channels. The subcellular localization of σ1 and its promiscuous receptor nature might explain this pleiotropic molecule’s wide range of cellular functions. The overexpression of IL-24 leads to ER stress due to impaired folding capacity and subsequent accumulation of misfolded proteins within the ER lumen. A stress response, the unfolded protein response (UPR), is induced to re-establish ER homeostasis. The interaction of misfolded proteins in the ER with BiP/GRP78 causes its dissociation from several effectors of the UPR that ultimately mediate suppression of protein translation and send misfolded proteins to degradation via the ER-associated degradation (ERAD) pathway. When the ER stress response is prolonged, apoptosis is induced by the transactivation of DNA damage-inducible transcript 3 (DDIT3, also known as C/EBP homologous protein, CHOP) expression. Evidence shows that members of the IL-20 subfamily, besides IL-24, trigger ER stress. For example, in colonic epithelial cells, IL-22, another interleukin from the same family of IL-24, induces an ER stress response [51] that may mediate pathogenicity in chronic colitis. 

## 8. Grim 19—Role in Mitochondria Autoimmune-Associated Pathways

Grim-19 (gene associated with retinoid-IFN-induced mortality 19) is upregulated after treatment with IFN-β and retinoic acid and overexpression of Grim-19 induce apoptosis [52,53]. Grim-19 is involved in apoptosis in cancer cells triggered by IFN-β and retinoic acid Grim-19. Grim-19 is also involved in mitochondrial metabolism and facilitates the translocation of STAT3 into the mitochondria [54]. Korn et al. demonstrated that Grim-19 interacts with IL-24 in the inner mitochondrial membrane, which contributes to the restriction of immunopathology in experimental autoimmune encephalomyelitis by promoting the accumulation of STAT3 in the mitochondria [7]. Therefore, future experiments must test if IL-24 acts through canonical or noncanonical pathways and whether those pathways are integrated or not. There is no evidence that other members of the IL-10 family, besides IL-24, interact with Grim-19.

## 9. Role in Lipid Homeostasis

The sphingolipid ceramide is part of a bioactive membrane lipid family implicated in mediating or regulating many diverse cellular processes, including cell cycle arrest, apoptosis, senescence, and stress responses. We and others have shown that induction of ceramide production plays a decisive role in IL-24-mediated apoptosis, with inhibition of ceramide production suppressing IL-24-induced apoptosis in several tumor cell lines [45,55,56]. Cellular ceramide is generated in response to stress pathways involved in apoptosis signaling, including treatment with prodeath ligands, anticancer chemotherapeutic drug treatments, and proinflammatory cytokines (e.g., TNFα, IL-1β). The TNF family of death receptors induces caspase-dependent programmed cell death through ceramide production; ceramide directly participates in apoptosis by forming large protein-permeable channels that enable the release from mitochondria of cytochrome c, which triggers downstream caspases. To our knowledge, IL-24 is the only cytokine from the IL-10 family able to induce ceramide production in cancer cells. Moreover, IL-10 decreases ceramide synthesis induced by TNFα in endothelial cells [57]. IL-24, by triggering ER stress, increases ceramide levels in cancer cells via PERK activity [58]. It is plausible that IL-24, by inducing TNFα and IL-1β, might also increase ceramide levels in certain cells [59,60]. 

## 10. PKA Pathway

Protein kinase A (PKA) activity depends on cAMP concentration, thus translating and integrating many signals that affect cAMP levels [61]. This signaling mechanism regulates myriad cellular functions and physiological and pathological processes. Among them PKA’s role in regulating IL-24-induced apoptosis, as we have demonstrated. We demonstrated that PKA activates the expression and activity of ATF4, the activator of TP53, and PKA is a mediator of extrinsic apoptosis in IL-24-mediated apoptosis [46]. PKA has been recognized as a potent negative regulator of ATM activation through PKA-dependent activation of PP2A in lung cancer cells [62]. IL-24 Inhibits lung cancer growth by suppressing glioma-associated oncogene homolog 1 (GLI1), associated with marked suppression of the ATM-mediated DNA damage response pathway, which results in increased DNA damage [63]. We can speculate that IL-24 inhibits ATM by PKA activation and GLI1 downregulation.

## 11. Role in Translation Regulation

The initiation stage is a key point in the regulation of protein translation. The main molecular players in this regulation are the translation initiation factors (eIFs), such as the ternary and eIF4F complexes [64,65,66]. Two major regulatory branches of translation initiation are the ternary and eIF4F complexes. Secondary structures in the mRNA untranslated regions, determined by their sequences, are important not only in binding to ribosomes but also to these translation initiation factors. The eIF2 trimeric protein complex (comprised of eIF2α, eIF2β, and eIF2γ) is essential for the formation of the ternary complex, a critical site of translational regulation. Phosphorylation of eIF2α on Ser51 restricts the formation of the ternary complex and is rate-limiting for translation initiation. We have shown that the anticancer effect of IL-24 is mediated by inhibition of translation initiation through eIF2α phosphorylation [47] and inhibition of the eIF4F complex [47,67], subsequently downregulating the synthesis of oncogenic proteins and reducing the overall rate of protein synthesis while preferentially upregulating the expression of tumor-suppressor and proapoptotic proteins [47,67]. In light of the tight translational control of proteins that promote cell growth and drive malignant transformation, the deregulation of translation initiation is recognized as an essential molecular mechanism underlying many human cancers. 

## 12. Conclusions

Since the discovery of IL-24 in the mid-1990s [68] and of its receptors in 2002 [10], we have learned the different functions of IL-24 on a broad array of target cells due to different signal transduction pathways that IL-24 can regulate. We also learned of the different spliced isoforms of IL-24, the interactions of IL-24 with receptors in the cell membrane, and its ability to interact with proteins in the ER, cytoplasm, and mitochondria [29]. To better understand the diverse effects of IL-24 on different cell types due to different signal transduction pathways, we must recognize the critical information gap remaining in our knowledge. Specifically, new research has shown the localization of IL-24 not only as a classical secreted interleukin binding to receptors in the plasma membrane and subsequently triggering JAK/STAT activation but also playing a role when it is localized in the endoplasmic reticulum, mitochondria, and cytoplasm (Figure 5). We were the first to demonstrate that IL-24 triggers apoptosis when it is not secreted but localized in the endoplasmic reticulum and independent of JAK/STAT pathways [15,17]. Furthermore, using different sublocalization IL-24 constructs, Sieger et al. demonstrated that only full-length or reticulum endoplasmic-targeted IL-24 transiently transfected constructs induce apoptosis in H1299 cells, but nuclear- or cytoplasmic-targeted IL-24 lack the capacity to trigger apoptosis [38]. The interaction of IL-24 with σ1 in the endoplasmic reticulum and the induction of ER stress are essential for cancer-specific apoptosis [12]. Very recently, Davidson et al. identified a shorter IL-24 variant isoform interacting with PKR; this interaction occurs in the cytoplasm and initiates a PKR-driven inflammatory response resulting from proteotoxic stress [9] (Figure 5). Allen et al. identified a novel splice variant of IL-24 that encodes a protein of 63 residues that lacks exons 3 and 5 of the full-length transcript from normal human melanocytes. This splice variant contains only 14 amino acids of homology to IL-24 located within the signal peptide region of the typical sequence of IL-24 that encodes a protein of 206 amino acids [69]. The fact that this IL-24 splice variant contains no predicted cleavage sites suggests it is not secreted. In normal human melanocytes, the IL-24 splice variant coexpressed with IL-24 full-length decreased the secretion of full-length IL-24. A specific IL-24 isoform that can block the secretion of IL-24 could be used as a therapeutic strategy. Five alternatively spliced isoforms of IL-24 induce apoptosis in cancer cells [70]. At the same time, an IL-24 splicing variant interacts with full-length IL-24 in the ER and inhibits secretion, blocking the antiapoptotic effect of full-length IL-24 [71]. Further experiments need to be conducted to determine the role of alternative splicing in the function of IL-24 related to the interaction with different IL-24 partners and the function and localization of the different isoforms. This could shed light on how IL-24 functions on a broad range of cell types (Figure 5). One of the challenges that we recognize is the complexity that exists in the interaction of multiple signals, in addition to IL-24, that will determine a particular response in a particular cell type based on the net signal triggered by the crosstalk of different inputs. These unexplored aspects will be essential to optimize the use of IL-24 as a therapeutic agent.

Although it is clear that IL-24 induces apoptosis in various cancer cells via JAK/STAT-independent pathways, further experiments need to be conducted to determine if, in some cases, IL-24 induces apoptosis in cancer cells by inhibiting JAK/STAT pathways through feedback inhibitors such as SOCS3 and Grim-19. For example, in memory B cells, IL-24 inhibits plasma cell differentiation by inhibiting the phosphorylation of STAT3 on tyrosine 705, followed by inhibition of the transcription of IL-10 and the inhibition of the transcription and activation of factors known to be involved in plasma cell versus memory B-cell commitment [72]. Moreover, IL-24 induces apoptosis through dephosphorylation of STAT3 and stabilization of p53 expression in chronic lymphocytic leukemia B cells [73]. 

We have demonstrated that IL-24 protein induces stabilization of its mRNA without activating its promoter [29]; it is possible that endogenous activation of IL-24 under inflammation can induce a robust expression of endogenous IL-24 and subsequent induction of ER stress-mediated pathways during immune regulation, tissue homeostasis, or host defense.

We and others have already demonstrated the therapeutic utility of IL-24, particularly, but not limited to, anticancer therapy. However, it is still necessary to fully understand the many actions of IL-24 in vivo and find ways to boost or block these mechanisms to attain the best therapeutic strategies.

## Figures and Tables

**Figure 1 cancers-15-03365-f001:**
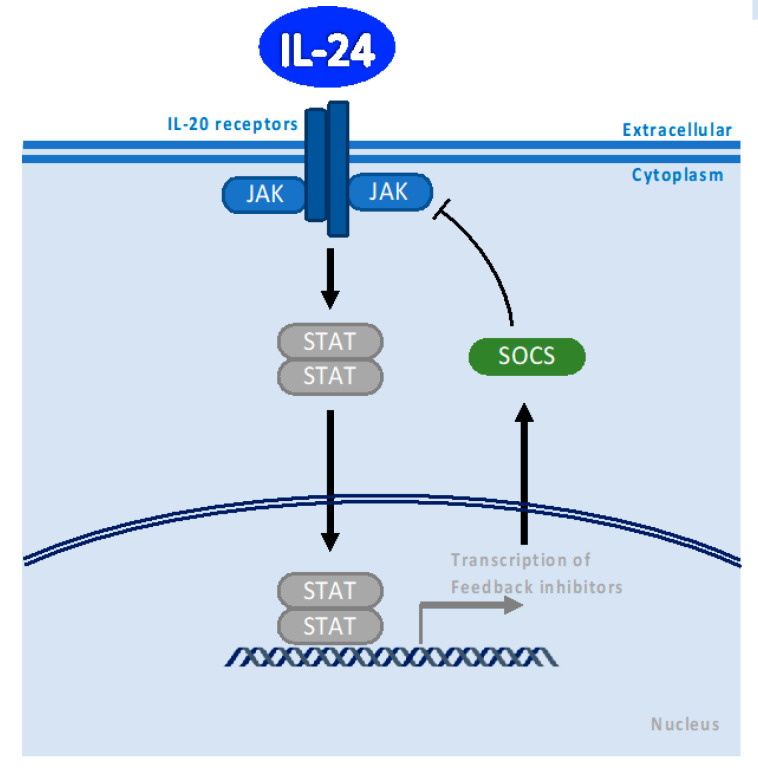
Model illustrating the canonical IL-24 signaling through binding to IL-20 receptor and JAK/STAT pathway activation and its regulation by SOCS proteins. The canonical IL-24 signaling is activated during Inflammation and tissue repair. Black arrows indicate pathway activation and black bar-headed arrow indicates inhibition.

**Figure 2 cancers-15-03365-f002:**
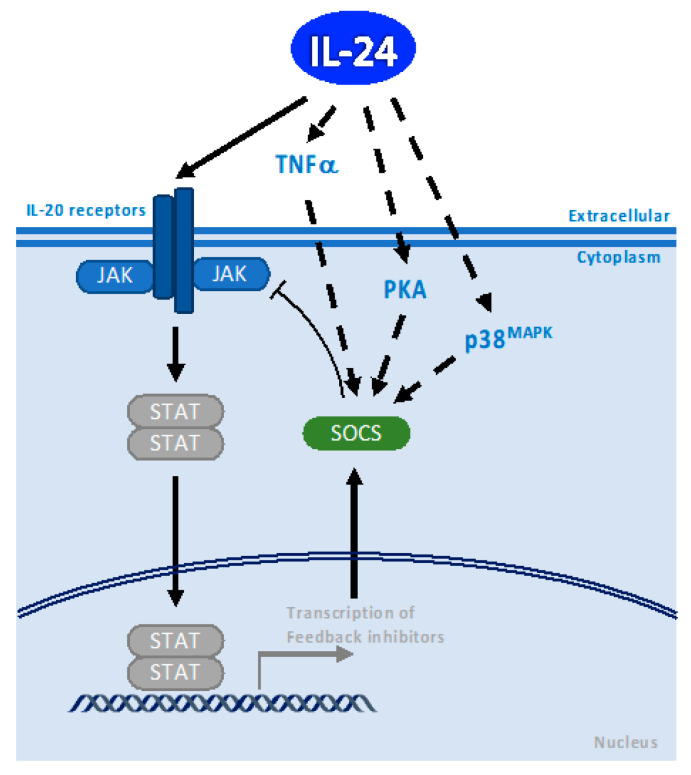
Model illustrating the possible molecular mechanism of SOCS activation by IL-24. IL-24 activates SOCS during Inflammation, tissue repair, and induction of cancer-specific cell death. Black arrows indicate pathway activation and black bar-headed arrow indicates inhibition. Black dotted arrows indicate a potential mechanism of action.

**Figure 3 cancers-15-03365-f003:**
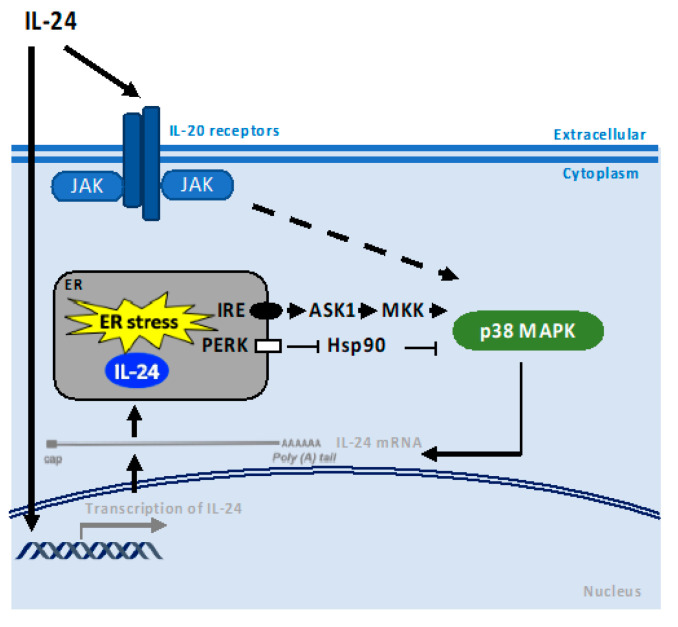
Model illustrating the possible molecular mechanism of p38 MAPK activation by IL-24. Adenovirus vectors (Ad) expressing secretable or nonsecretable IL-24 (Ad.IL-24), as well as IL-24 protein (generated from Ad.IL-24-infected cells), all display broad cancer-specific proapoptotic activity through induction of endoplasmic reticulum (ER) stress. Both IL-20 receptor and ER stress lead to p38 MAPK activation after IL-24 treatment. p38 MAPK regulates IL-24 expression by stabilization of the 3′UTR. Black arrows indicate pathway activation and black bar-headed arrow indicates inhibition. Black dotted arrows indicate a potential mechanism of action.

**Figure 4 cancers-15-03365-f004:**
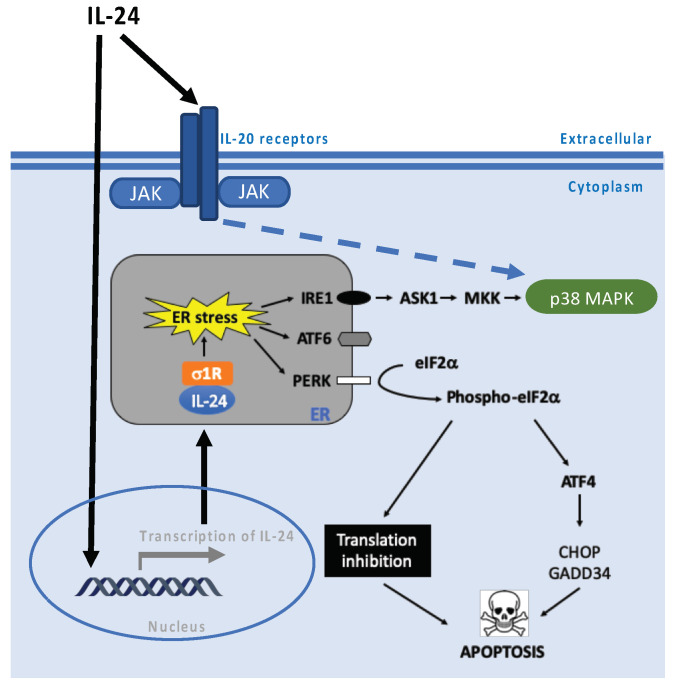
Model for the mechanism of induction of cancer-specific cell death triggered by IL-24. Black arrows indicate pathway activation and black bar-headed arrow indicates inhibition. Black dotted arrows indicate a potential mechanism of action.

**Figure 5 cancers-15-03365-f005:**
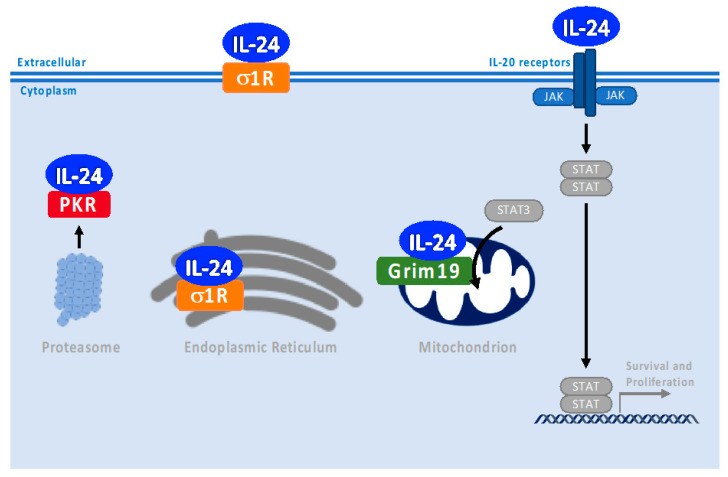
Summary of canonical and noncanonical signaling pathways of IL-24 and its molecule partners. IL-24 can bind not only to its classic protein plasma membrane receptors, the IL-20 Receptors, but also to other intracellular molecular partners, including σ1, Grim19, and PKR. Starting from the left: IL-24/PKR-driven inflammatory response resulting from proteotoxic stress; IL-24-induced cancer-specific apoptosis by interaction with σ1 in the endoplasmic reticulum and plasma membrane; IL-24 interacts with GRIM19, a respiratory chain protein in the inner mitochondrial membrane in T cells, promoting the recruitment of STAT3 to mitochondria; and lastly, the canonical signaling pathways of IL-24 dependent on the interaction with IL-20 receptors on the plasma membrane during inflammation, host defense, immune regulation, and tissue repair.

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
