# Peer review of "Interleukin 24: Signal Transduction Pathways"

_cancers, 2023, doi:10.3390/cancers15133365_

Round 1

Reviewer 1 Report

The authors have summarized the signaling pathway of Il24 in different context. Though the Figures shown by the authors are simple, but fails to convey the importance of cross talk between the pathways; as it is know, that IL signaling intermediates are not just activated by IL24 alone, but various others as well; so it is important to discuss if there is signaling exclusivity in certain cell types or in pathophysiology; further the MS lacks details about the clinical relevance; it is important to summarize in a Table, about the expression levels of IL24 & its receptor in various cancer types or other disease conditions  how was it determined, sample size & sample cohort, etc. This information will enrich the review & will definitely benefit the readers & explicitly state the importance of targeting IL24 signaling pathway for therapy purpose. The final mechanism figure 5 lacks a wholistic picture and how these intra cellular events are connected; in Figure 5, 3 distinct intra cellular compartments are shown, but how are they linked to IL24 signaling; are there any unknown/less studied pathways linking them or known intermediates exist?

Author Response

We thank the Reviewer for carefully reviewing the manuscript and his or her generous comment about the originality of the work and the potential importance of the findings.

  1. In response to the Reviewer’s request to emphasize the importance of cross-talk between the pathways. Following the Reviewer's comment, the following paragraph has been added to the discussion: “One of the challenges that we recognize is the complexity that exists in the interaction of multiple signals, in addition to IL-24, that will determine a particular response in a particular cell type based on the net signal triggered by the crosstalk of different inputs. These unexplored aspects will be essential to optimize the use of IL-24 as a therapeutic agent.”

  1. As the Reviewer suggested, we added information about expression levels of IL-24 and receptors in cancer types. Following the Reviewer’s comment, the following paragraph has been added: “Although the expression of IL-24 receptors is limited to specific tissues, in cancer cells, the expression of IL-24 receptors is widely expressed6, including in melanoma, prostate, pancreatic, fibrosarcoma, ovarian, and breast cancer.”

  1. In response to Reviewer’s request about Figure 5, the following paragraph has been added to the Figure 5 legend: “Summary of canonical and non-canonical signaling pathways of IL-24 and its molecule partners. IL-24 can bind not only to its classic protein plasma membrane receptors, the IL-20 Receptors, but also to other intracellular molecular partners, including s1, Grim19, and PKR. Shown, starting from the left, IL-24/PKR-driven inflammatory response resulting from proteo-toxic stress; IL-24-induced cancer-specific apoptosis by interaction with s1 in the endoplasmic reticulum and plasma membrane; IL-24 interacts with GRIM19, a respiratory chain protein in the inner mitochondrial membrane in T cells, promoting the recruitment of STAT3 to mitochondria; and lastly, the canonical signaling pathways of IL-24 dependent on the interaction with IL-20 receptors on the plasma membrane during inflammation, host defense, immune regulation, and tissue repair.”

Reviewer 2 Report

The authors present a concise and informative review on IL-24, which undoubtedly contributes to promoting the interest and understanding of this yet-to-be explored member of the cytokine kingdom.

 In particular, the JAK-STAT independent signaling by this cytokine which seems to play an important role in regulating cancer growth is intriguing, as this aspect is not shared by many other cytokines.

 This is a comprehensive and well-written manuscript so there are only minor concerns by this reviewer, as follows.

 The secretion of IL-24 is interesting, to an extent similar to that of interleukin 15, which also has a long signal peptide of 48 aa. It is logical to assume that the intracellular trafficking of IL-24 is very complicated. While it is intriguing that the intracellular version of this cytokine seems equally active as the secretable form, whether or not this reflects a physiological state of IL-24/MDA-7 was not shown in the original publication (ref 17) so this reviewer wonders if any evidence has been even shown that cells physiologically expressing MDA-7/IL-24 display similar intracellular retention of this molecule as that in cells transfected with SP-less IL-24? If so, please show the reference as this observation seems to be made only by the authors’ group so far, but not verified by other groups.

Chaperon -ER stress / Grim 19

These appear to be unique features associated with IL-24. Are these shared with other IL-10 family members? Or as mentioned about lipid homeostasis, only attributable to IL-24? If so, perhaps it might be worth mentioning it.

Author Response

We thank the Reviewer for carefully reviewing the manuscript and his or her generous comment about the originality of the work and the potential importance of the findings.

  1. In response to Reviewer 1 request to clarify IL-24 displays similar intracellular retention of this molecule as that in cells transfected with SP-less IL-24 by other than the authors’ group, the following paragraph has been added to the discussion: “Furthermore, using different sub-localization IL-24 constructs, Sieger et al. demonstrated that only full-length or reticulum endoplasmic-targeted IL-24 transiently transfected constructs induce apoptosis in H1299 cells, but nuclear- or cytoplasmic-targeted IL-24 lack the capacity to trigger apoptosis38.”
  2. As the Reviewer suggested, we added information about the only attribute of IL-24 associated with interaction with the Grim-19 protein. The following paragraph has been added to the manuscript: “There is no evidence that other members of the IL-10 family, besides IL-24, interact with Grim-19.”

Round 2

Reviewer 1 Report

Recommended.